# Multiplex Point-of-Care Tests for the Determination of Antibodies after Acellular Pertussis Vaccination

**DOI:** 10.3390/diagnostics10040187

**Published:** 2020-03-27

**Authors:** Aapo Knuutila, Carita Rautanen, Jussi Mertsola, Qiushui He

**Affiliations:** 1Institute of Biomedicine, University of Turku, Kiinamyllynkatu 10, 20520 Turku, Finland; aajukn@utu.fi; 2Department of Biotechnology, University of Turku, Kiinamyllynkatu 10, 20520 Turku, Finland; carita.a.rautanen@utu.fi; 3Department of Pediatrics and Adolescent Medicine, Turku University Hospital, Kiinamyllynkatu 4-8, 20520 Turku, Finland; jusmer@utu.fi; 4Department of Medical Microbiology, Capital Medical University, No.10 Xi Tou Tiao, You’an Men Wai, Feng Tai District, Beijing 100069, China

**Keywords:** pertussis, in vitro-diagnostics, point-of-care, serology, lateral flow, multiplex

## Abstract

Most of the current serological diagnosis of pertussis is based on pertussis toxin (PT) IgG antibodies and does not differentiate between vaccination and infection-induced antibodies. PT is included in all of acellular pertussis vaccines available in the world. Multiplex testing of non-vaccine antigen-related antibodies has the potential to improve the diagnostic outcome of these assays. In this study, we developed a quantitatively spatial multiplex lateral flow immunoassay (LFIA) for the detection of IgG antibodies directed against PT, pertactin (PRN), and filamentous hemagglutinin (FHA). The assay was evaluated with serum samples with varying anti-PT, anti-PRN, and anti-FHA IgG levels and the result was compared to those obtained with standardized ELISA. The developed assay showed good specificity with PT and PRN antibodies and semiquantification throughout the antigen combinations. This exploratory study indicates that the multiplex LFIA is specific and sensitive, and a similar test platform with alternative antigens could be suitable for new type of pertussis serology.

## 1. Introduction

Laboratory diagnosis of pertussis infections, caused by respiratory pathogen *Bordetella pertussis*, is important for the surveillance, treatment, and prevention of the disease. The diagnosis of pertussis infections in the early stage is based on culture and PCR, and in the late stage on serology. Serological analysis by ELISA has been widely used for the evaluation of antibody responses to pertussis vaccination and infection and for serosurveillance [1,2]. Single high values of IgG and IgA antibodies to pertussis toxin (PT) usually indicate infection in individuals older than two years of age [3]. Furthermore, in single-point serology, the measurement of IgG antibodies against PT is recommended, since it has a higher sensitivity than anti-PT IgA antibodies [4]. The serological single point diagnostics based on PT is mainly interfered with vaccine-induced antibodies, as separating vaccine-induced anti-PT IgG responses from a recent pertussis infection is especially difficult after six months and even up to two years after the latest vaccination [5]. Besides vaccinations, *B. pertussis* is also continuously circulating in the population, making it challenging at times to separate recent and earlier infections, which drives the need for paired serum samples collected in the early and late stages of the disease, and high cutoffs for positive diagnosis [6]. Older children and adults especially often have delayed suspicion of pertussis, and thus early samples for paired serology are difficult to obtain.

To combat some of these challenges, multiplex serological assays to *B. pertussis*-specific antigens could improve diagnostics in comparison to single antigen-based IgG/IgA measurements, but in return complicate and increase the workload required to attain a diagnosis. Multiplexing may prove useful, especially as the immune response to pertussis vaccination and infection varies between individuals and *B. pertussis* organisms. Pertussis caused by certain antigen-deficient strains have already been reported and result in a negative antibody response to such antigens. At present, this is mainly a problem with pertactin (PRN) and not with PT [7,8,9,10]. Although purified PT is the recommended antigen for ELISA, combinations with common antigens filamentous hemagglutinin (FHA) and PRN may improve its diagnostic sensitivity [11]. However, these two antigens also have the same underlying challenges as vaccine-included antigens, and cross-reacting antibodies to these antigens are also induced by other bacteria. For optimal serological diagnosis of pertussis, antigens which are not included in current acellular vaccines—such as adenylate cyclase toxin (ACT), fimbriae 2/3 and virulence-associated gene 8 (Vag8)—could be considered [12,13,14,15]. Therefore, a combinatory multiplex test with well-established, quantitative cut-off values for antibodies against purified pertussis antigens in different combinations may offer a useful tool for rapid diagnosis and differentiation.

ELISAs are time-consuming and laborious, and pose limitations when the volume of serum samples may be limited when required to quantify the immune response to pathogens presenting multiple virulence factors. Cost-efficient and high-throughput multiplex serological assays in the pertussis field already have a broad application [16,17,18,19]. We have reported earlier on a simple, quantitative and rapid lateral flow (LF)-based platform for anti-PT IgG serological diagnostics of pertussis without the complexity of common laboratory practicalities [20]. The apparent advantages of multiplexing a serological response with LF immunoassays (LFIA) are related to the easily accomplished test setting through multiple test lines and a common label without any further complications either to design or to perform the test in comparison to single test line-based tests. In this study, we used the developed LF-platform for the multiplex measurement of antibody responses in acellular pertussis vaccine recipients to multiple pertussis antigens.

## 2. Materials and Methods

Serum samples and reference assays: A total of 19 samples (Table 1) with highly varying anti-PT, FHA and PRN antibody concentrations were selected from pre- and post-vaccination samples of Finnish children who received a booster dose of a three-component dTap vaccine, given at 11–13 years of age in 1997 [21]. The amounts of serum IgG antibodies to PT, FHA, and PRN had been previously measured by ELISA. In addition, in-house positive and negative controls were included for each antigen for initial testing.

Label conjugates: A total of 375 µg of Protein A (#RL240425, Thermo Fisher Scientific, Meridian Rd., Rockford, IL, USA) was conjugated with activated fluoro-Max carboxyl modified microparticles containing europium chelates (diameter 99 nm, Thermo Fisher Scientific), as described earlier [20]. Protein A has the ability to bind to anti-human IgA, IgG, and IgM simultaneously, and has been preferred in our assays due to its small size in comparison to larger secondary antibody conjugates [22,23].

Lateral flow test strips and multiplex assay: The preparation of the strips and the test procedure was performed following a similar method to Salminen et al. [20] with the following modifications: The captured reagents on the test strips were purified PRN, PT and FHA—kindly provided by GlaxoSmithKline (Belgium), and rabbit anti-goat IgG antibody on the control line (Figure 1). Lateral flow test strips were assembled on a plastic support (G&L Precision Die Cutting, San Jose, CA, USA) with a cellulose absorbent pad (CFSP223000, Millipore, Bedford, MA, USA) and a glass fiber sample pad (G041, Millipore). Two hundred ng/cm of antigens in 10 mM Tris-HCl (pH 8.0) buffer was dispensed on the nitrocellulose membrane (Hi-Flow Plus HF90, Millipore) with Linomat 5 (Camag AG, Muttenz, Switzerland) liquid dispenser. The control line was printed 8 mm from the FHA test line with 400 ng/cm of rabbit antibody in Tris-HCl.

The lateral flow assay was performed in two simple steps in less than 30 min. First, strips were dipped with serum diluted 1:180 in 60 μL of assay buffer (10 mM Na_2_HPO_4_, 135 mM NaCl, 1% BSA, 0.5% Tween-20), with two replicate strips for each sample. Second, after the sample had passed through the test, 2 × 10^7^ Protein-A conjugated Eu-nanoparticles in 60 µL of assay buffer was added. Time-resolved fluorescence (615 nm) was measured, for the consistency of the results, after two hours by line scanning the area of the strips containing the test and control lines with measurement points at 1.1 mm intervals with a Victor 3 multi-label plate reader (Perkin-Elmer, USA).

## 3. Results

A simple multiplex lateral flow point-of-care (POC) assay was developed for the detection of anti-PRN, anti-PT, and anti-FHA antibodies. Serum samples with highly variating antibody levels between the different test antigens were tested in the LFIA to demonstrate the specificity of the assay. The average coefficient of variation in the LFIA between two replicates was 11.1% when calculated from test line signals, which ranged between 9.6–12.3% between the different antigens. We do not encourage the use of ratiometric test-to-control line analysis in multiplex assays, as the control line signal fluctuates in regard to each test line. In Figure 2, the specificity of the assay is demonstrated between zero, one, two or three possible positive test results. Excellent differentiation was achieved over all of the combinations, except in anti-FHA IgG-positive samples, which had clear cross-reactivity with the PT test line (Figure 2d,g). This was likely due to the unspecific binding of the protein-A label to the IgA and IgM antibodies from the samples, as a test with anti-human IgG label conjugate (Figure 3) produced no signal with the same samples from the PT test line.

## 4. Discussion

In this work, we used purified PT, FHA, and PRN antigens that are included in the current acellular pertussis vaccines to explore the possibility of spatially multiplexing up to three serological responses in a single lateral flow test. The order of the test lines on the strips was chosen based on the molecular weight of the antigens from the smallest to the largest. Arguably, whichever the order, the flow of the label was not affected, but hypothetically, the use of FHA as the first line should be avoided to ensure a stable flow of the sample. It was particularly clear from the overall signal profiles of triple-antigen positive samples that a very high signal from the first lines decreases the signal intensity of the following lines and the background to an extent. Therefore, the signal of the FHA-test line was easily saturated, as was observed from samples with FHA concentrations ranging from 481 to 4205 EU/mL, that gave near-equal signals in the LF assay (Table 1, Figure 2a). Overall, the readout remains a clear peak, and in these cases, the readout can be considered as at least semi-quantitative. The same phenomenon of signal saturation was quite consistently visible in the control lines in most of the positive samples; and when a negative sample was used, the control line peaks were generally the highest. All things considered, the results showed both good specificity and quantitativity, especially across PT and PRN test lines.

As current anti-PT IgG assays—or anti-PRN and anti-FHA for that matter—do not differentiate between antibodies induced by acellular vaccines and infection, multiplex immunoassay using different combinations of other pertussis relevant antigens simultaneously may offer a useful tool for rapid diagnosis and differentiation. Even though all of the tested antigens are included in most of the current acellular pertussis vaccines, the results are promising for the further development and evaluation of rapid multiplex testing up to three different serological correlates. At present, we are testing ACT, an antigen that is not included in the current acelluar pertussis vaccines in different comibinations with the above-mentioned antigens.

In addition to antigen-based multiplexing, other possibilities in serology include the determination of antibody isotypes and subclass. Ig isotypes appear at different times after the onset of disease, [24], and therefore IgA and IgM can be considered as more specific to acute infections in comparison to IgG—on top, vaccinations mainly produce IgG [3,25,26]. Additionally, the simultaneous measurement of IgA, IgG and IgM antibody levels has been reported to provide the most diagnostic accuracy [27], especially if interpreted in connection with clinical findings [28], but may also lead to overdiagnosis [14,29]. Considering IgG subclasses, IgG1 and 4 responses were elevated to multiple antigens in children primed with acellular pertussis vaccines [17,30,31,32], whereas, in patients or unvaccinated convalescent children, IgG1 and IgG3 were more prevalent [30,31,33]. Therefore, the multiplexing of IgG1, 3 and 4 could provide a more comprehensive image to separate recent vaccination and infection. In our experience, however, multiplexing with Ig isotypes or subclasses is challenging in LFIAs. Instead of spatial multiplexing, spectral multiplexing—with, e.g., fluorescent labels—may prove to be more feasible to accomplish [34]. 

Although not a primary aim of the developed assay, the used protein A label in itself provides a possible improvement in the sensitivity and timing of rapid assays in comparison to a sole IgG measurement; however, the samples selected for this study do not strictly show evidence for it. Interestingly, the unspecific binding of protein A to PT test line was found only in anti-FHA (/+PRN) IgG-positive serum samples (Figure 2d,g), whereas other samples did not show any cross-reaction with the other test lines. Only very small anti-PT IgA and IgM antibody concentrations were found by ELISA from these samples, which would not explain such a high signal response in the LFA. Other causes are hard to outline, as no unspecific binding was noticed after a switch to an IgG-specific label (Figure 3). Further use and research of multiple isotype binding protein labels are needed from early acute phase patients to better evaluate whether the approach of measuring overall antigen-specific Ig-levels could improve sensitivity in a similar way to Ig isotype multiplexing [27]. The relative proportions of IgM/IgA/IgG and their influence on the signal may, in the end, turn out to be difficult to interpret.

The study has several limitations. First, the number of samples in this study was small, and were chosen from vaccine recipients. Second, the samples were chosen manipulatively and would be very rare to come by in a clinical setting (roughly a chance of one in twenty in the sample material of the chosen vaccination study). The samples were selected, above all, to demonstrate specificity, and to an extent quantitative aspects. Therefore, the study samples do not by any means reflect the reality of infections, in which antibody responses would most often be positive towards all antigens. In addition, these sera used have been stored for 20 years. Although they have been stored without freezing and thawing, the possibility that degradation of antibodies may influence the results could not be excluded. The extent of this phenomenon and the overall design of the test to have the FHA test line last was visible, especially when comparing the weakest quantification of the FHA test line to the other test lines.

In conclusion, this exploratory study indicates that the developed multiplex LFIA can be a suitable option for the point-of-care measurement of antibodies against pertussis. Compared to ELISAs and other multiplex formats, the LFIA offers a very rapid readout, a low amount of antigens and samples, and ease of use [20]. The specificity of the spatially multiplexing design gives an encouraging glance for future work especially with antigens, which are not included in the current acellular vaccines. Further study with a set of sera collected from patients with laboratory-confirmed pertussis is needed, together with the consideration of novel approaches for the rapid multiplexing of overall antibody subclasses and isotypes.

## Figures and Tables

**Figure 1 diagnostics-10-00187-f001:**
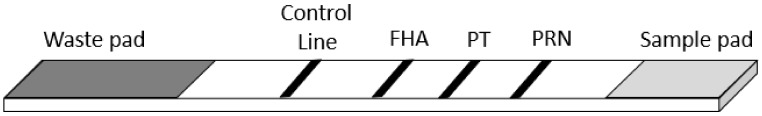
The layout of the multiplex lateral flow test. Pertactin (PRN), pertussis toxin (PT) and filamentous hemagglutinin (FHA) antigen test lines were absorbed on the nitrocellulose membrane, in respective order.

**Figure 2 diagnostics-10-00187-f002:**
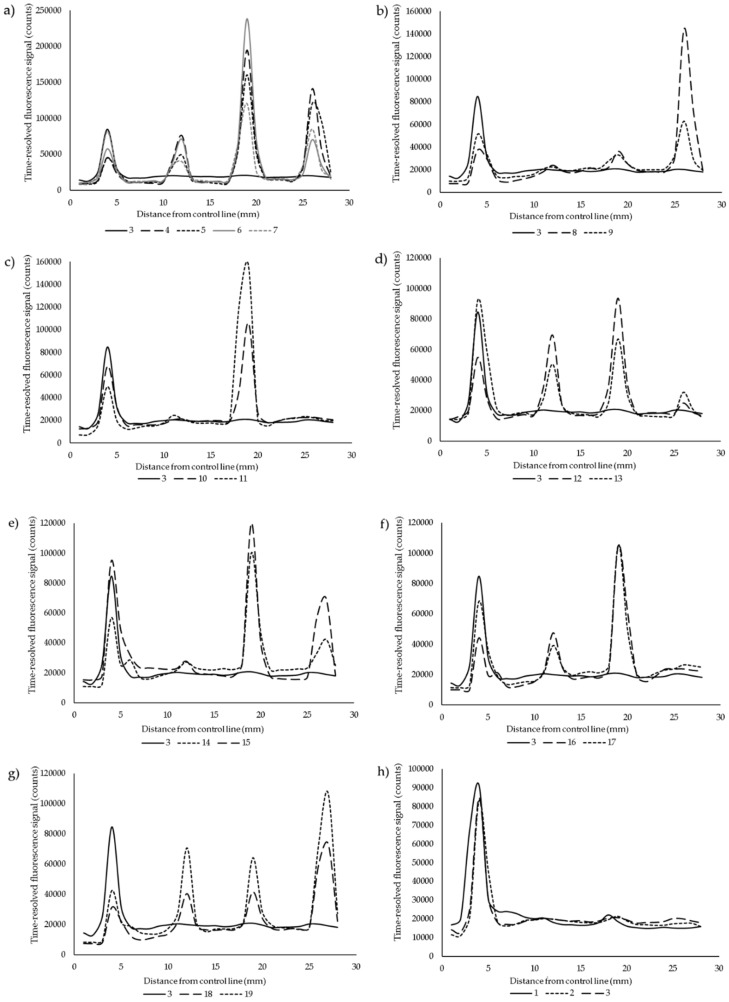
The multiplex readouts from lateral flow test strips were measured as average time-resolved fluorescence signal by line scanning the nitrocellulose membrane starting from the control line (4 mm), followed by FHA (12 mm), PT (19 mm) and PRN (27 mm) test lines. Different combinations of anti-PRN, -PT and -FHA-specific IgG antibodies were tested: (**a**) All positive (samples 4, 5, 6, 7), (**b**) only PRN IgG positive (8, 9), (**c**) only PT positive (10, 11), (**d**) only FHA positive (12, 13), (**e**) both PRN and PT positive (14, 15), (**f**) both PT and FHA positive (16, 17), (**g**) both PRN and FHA positive, (18, 19), and (**h**) all negative samples (1, 2, 3). Please refer to Table 1 for antibody concentrations.

**Figure 3 diagnostics-10-00187-f003:**
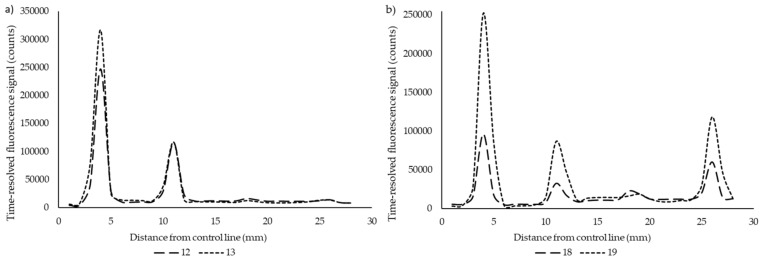
A test with anti-human IgG label demonstrated no unspecific binding to PT test line in (**a**) only anti-FHA (12, 13) and (**b**) both anti-FHA and anti-PRN (18, 19) IgG positive serum samples.

**Table 1 diagnostics-10-00187-t001:** Study samples with known concentrations of anti-PT, –PRN and –FHA IgG antibodies (ELISA units/mL).

Sample	Anti-FHA IgG	Anti-PT IgG	Anti-PRN IgG
1	2.5	2.5	2.5
2	2.5	2.5	2.5
3	2.5	2.5	2.5
4	4205	582	1592
5	1364	580	1339
6	1730	578	314
7	481	168	466
8	66	2.5	948
9	63	2.5	230
10	14	68	6
11	47	343	11
12	1280	17	55
13	370	2.5	64
14	66	145	119
15	67	92	255
16	400	79	2.5
17	162	84	6
18	440	2.5	574
19	815	6	1652

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
