# Peer review of "Multiplex Point-of-Care Tests for the Determination of Antibodies after Acellular Pertussis Vaccination"

_diagnostics, 2020, doi:10.3390/diagnostics10040187_

Round 1
Reviewer 1 Report
Fast and accurate diagnosis of any infectious disease, including pertussis if of high importance. Even though there are PCR- and antibody-based assays, there are situations showing weaknesses of these assays (e.g. interference form post-vaccination antibodies). The Authors propose new approach to the problem that sound like having high potential in accurate diagnosis of Bordetella pertussis infection.
However, as the Authors state themselves, the samples were collected only from vaccine recipients (lines 182-183) and this fact, should be emphasized in introduction. Perhaps the Authors might consider the change of the title stating that there are preliminary results in terms of using the tests in the case of real infection?
Figure 2 is missing in the manuscript. Please, include it in the revised version of the manuscript.
Author Response
We appreciate these comments to our manuscript.
Reviewer 1
1) "the samples were collected only from vaccine recipients (lines 182-183) and this fact, should be emphasized in introduction." & 2) "Authors might consider the change of the title stating that there are preliminary results in terms of using the tests in the case of real infection?"
Response: The following changes were made to address these comments:
New title: Multiplex point-of-care tests for determination of antibodies after acellular pertussis vaccination
"In this study, we used the developed LF-platform for the multiplex measurement of antibody responses from acellular pertussis vaccine recipients to multiple pertussis antigens." (Lines 71-73)
3) Figure 2 is missing in the manuscript. Please, include it in the revised version of the manuscript.
Response: It seems that some system in the submission process did not support the Figure 2. file type within the word file and was removed. It should now be (once more) visible within the submission file and as a seperate image file.
Reviewer 2 Report
The authors present a manuscript on the development of a lateral flow approach to antibodies for pertussis.
All sections of the manuscript are well written and the work looks to have been carried out to a high standard In many ways I would like to approve this article for publication.
However I found the results section and the number of data plots presented to be thin. A lost of aspects are covered in the results and discussion section and I could not find the corresponding data to corroborate what was being discussed. With the data presented being thin it is very hard to gain an accurate impression over the quality of this work.
I would encourage the authors to think about including more data and pointing more clearly to it when discussing the performance, advantages and limitations of their work.
Author Response
We appreciate these comments to our manuscript.
Reviewer 2
1) "However I found the results section and the number of data plots presented to be thin. A lot of aspects are covered in the results and discussion section and I could not find the corresponding data to corroborate what was being discussed. With the data presented being thin it is very hard to gain an accurate impression over the quality of this work."
& 2) "I would encourage the authors to think about including more data and pointing more clearly to it when discussing the performance, advantages and limitations of their work"
Response: First, we hope that the correct inclusion of Figure 2 will improve this impression.
Following changes were made:
Added multiple references to Figure 2&3 in the discussion section to point more clearly the implications of data.
"unspecific binding of protein A to PT test line was found only in anti-FHA (/+PRN) IgG positive serum samples (Figure 2 d&g), " (Lines 176-7)
Other causes are hard to outline, as no unspecific binding was noticed after a switch to an IgG-specific label (Figure 3). (Lines 180-1).
It is particularly clear from the overall signal profiles of triple-antigen positive samples that a very high signal from the first lines decreases the signal intensity of the following lines and the background to an extent. Therefore, the signal of the FHA-test line was easily saturated, as was observed from samples with FHA concentrations ranging from 481 up to 4205 EU/ml that gave near equal signal in the LF assay (Table 1, Figure 2 a).” (Lines 139-143)
It should also be clearer with the following addition to outline when all samples (graphs) should be considered:
The same phenomena of signal saturation was quite consistently visible in the control lines in most of the positive samples; and when a negative sample was used, the control line peaks were generally the highest. (Lines 145-147)
Third paragraph of Discussion was made more concise, as it has no corresponding data.
Rephrased the study limitation paragraph to be more clear. (Lines 189 forward)
Removed some repeating elements on the concluding paragraph.